# Tensile Behaviors and Mechanical Property Analyses of T-Welded Joint for Thin-Walled Parts in Consideration of Different TIG Welding Currents Using Multiple Damage Models and Fracture Criterions: Numerical Simulation and Experiment Validation

**DOI:** 10.3390/ma16134864

**Published:** 2023-07-06

**Authors:** Minghui Pan, Yuchao Li, Siyuan Sun, Wenhe Liao, Yan Xing, Wencheng Tang

**Affiliations:** 1School of Mechanical Engineering, Nanjing University of Science and Technology, Nanjing 210094, China; lyc1250477593@163.com (Y.L.); 18852724510@163.com (S.S.); cnwho@njust.edu.cn (W.L.); 2Digital Forming Technology and Equipment National-Local United Engineering Laboratory, Nanjing University of Science and Technology, Nanjing 210094, China; 3School of Mechanical Engineering, Southeast University, Nanjing 211189, China; xingyan@seu.edu.cn (Y.X.); tangwc@seu.edu.cn (W.T.)

**Keywords:** T-welded joint, welding current, tensile property, fracture criterion, finite element simulation

## Abstract

In order to deeply investigate the tensile properties and fracture behaviors that are obtained by tensile tests of welded joints, constitutive and damage models are imperative for analyzing the tensile behaviors. In this work, the tensile tests are conducted on the T-welded joint specimens of aluminum alloy 6061-T6, which were cut from the T-welded joints of thin-walled parts under different welding currents of Tungsten Inert Gas Welding (TIGW). A modified Johnson-Cook (J-C) model based on the original J-C equation, Swift model, Voce model, and Hockett-Sherby (H-S) model, their linear combination model, and fracture failure model are constructed and applied to simulate tensile behaviors, combined with tensile test data. What is more, the finite element (FE) simulation of tension tests is executed with the VUMAT and VUSDFLD subroutines. Compared to those results simulated with different fracture criteria and tensile experiments, the tensile strength and yield strength of T-welded joint thin-walled parts under different welding currents were achieved, and their best mean errors were only about 1%. Furthermore, the accuracy of different fracture criteria is also evaluated by the correlation coefficient and mean squared error. The results show that the combination model can accurately predict the tensile properties and fracture behaviors of T-welded joints better than the single model, especially the results simulated with the Swift-H-S model and H-S-Voce model, which are in good agreement with tensile test results, which will provide an analysis foundation for enhancing the welding assembly quality and preventing fracture failure for complex thin-walled antenna structures.

## 1. Introduction

Arc Welding assembly techniques have been widely applied in the thin-walled part structures and manufacturing of automobiles, aircraft, radar antennas, marine equipment, etc. [1,2,3]. The performance and reliability of the welding structure for the core component of that high-end equipment are significantly affected by the influence of welding techniques, welding materials for thin-walled parts, etc. Especially the welded joint of the welding structure is the weakest zone in the welding assembly, and the service behaviors and mechanical properties involving the ultimate strength are critically imperative for the thin-walled structure of the welding assembly [4,5]. Moreover, welding-induced deformation is a major concern in the assembly process caused by the non-uniform temperature field and rapid cooling, and the adverse effect of the welding deformation on the strength and fracture behaviors, ductile damage, assembly accuracy, stiffness distribution, vibration characteristics under different welding parameters, and service performance of welding structures does exist [6,7,8,9,10]. Therefore, the welding properties and performance of metal sheet parts are closely related to welding deformation. The desired weld quality, achievable strength, and joint soundness are also greatly influenced by the induced heat, material flow, and residual stress [11].

Based on this, the main performances involving mechanical properties, welded joint strength, and fracture failure behaviors can be characterized by the quantitative relationship between stress and strain, which is obtained by tensile testing and FE simulation. Heretofore, the stress-strain, tensile behaviors, and fracture mechanisms of metal sheet parts have been analyzed by a large number of researchers through some mathematical constitutive models and damage models. For example, considering different temperatures and strain rates, Lee et al. analyzed the mechanical response and microstructure evaluation of alloy aluminum 6061-T6 and investigated the flow stress-strain response under different loading conditions described with the Zerilli-Armstrong fcc model [12]. Roth et al. employed the J-C plasticity model combined with the Swift-Voce strain hardening law to depict the local strain fields of high-strength steel sheets [13]. Zhang et al. presented a novel approach based on the modified Voce model to analyze the flow stress and deformation behavior of AA5086 sheet at different temperatures and strain rates and analyzed the effect of temperature and strain rate on sheet formability through experimentation and numerical simulation [14]. Tan et al. studied the flow behaviors of 7050-T7451 aluminum alloy under different strain rates with the Johnson-Cook model combined with a correction to the strain rate hardening coefficient [15]. What is more, considering the stress triaxiality and size effect on damage evolution under different stress states, the relationship among the material damage and stress-strain damage parameters is also revealed with the Gurson-Tvergaard-Needleman (GTN) model through FE simulation [16,17]. Additionally, Abi-Akl et al. analyzed the anisotropic plasticity and the fracture initiation in pre-strained, artificially aged aluminum 6451 sheets and researched the material’s strain hardening behavior and the stress-state sensitivity with the Swift-Voce hardening model [18]. Cao et al. analyzed the working hardening behavior of different metallic alloys with Hollomon, Swift, and Voce models and compared the prediction accuracy of tensile strength for different hardening models [19]. Pham et al. analyzed the effect of the post-necking behaviors of the Kim-Tuan hardening model, the Swift model, the Voce model, the Hockett-Sherby (H-S) model, the Ghosh model, and a linear combination of the Swift and Voce models on the theoretical forming limit curve of aluminum alloy 5052-O and 6016-T4 sheets under punch-stretching tests [20]. Liu et al. investigated the microstructure evolution, mechanical properties, and fracture behavior of 6061-T6 thin plate joints considering welding speed for friction stir welded and analyzed the reasons for weld softening, which resulted in different fracture characteristics during the tensile process [21]. Yang et al. researched the material flow of joints in the friction welding process of 2198-T8 aluminum alloy with an arbitrary Lagrangian-Eulerian method, and the variation of flow stress and strain rate of TB9 alloy was analyzed using the J-C model with temperature [22,23]. Erice et al. adopted the Swift-Voce hardening and J-C model of strain rate and temperature dependency to analyze the deformation response, and the effect of stress state on strain to fracture for high-strength steel was analyzed with the Hosford-Coulomb model [24]. Wang et al. studied the tensile forming limit of the strip specimen with the GTN model and the effect of inclusions on matrix deformation and fracture behavior of the forged 304 stainless steel [25]. Zhang et al. researched the thermal history and mechanical properties of the friction stir spot welded joint of AA6061-T6 by combining microstructure-based modeling with thermo-mechanical coupling simulation and analyzing the tensile fracture behaviors of the joint model with the J-C failure law [26]. Jia et al. studied the material flow properties of AA 6016-T6 using uniaxial tension simulation under different strain rates and a modified constitutive J-C model to accurately predict the stress, strain, and plastic deformation behaviors [27]. Rotpai researched the flow stresses and hardening behaviors of aluminum alloy AA7075 under tensile tests for different temperatures through the Kocks-Mecking and Crussard-Jaoul models and the Hollomon, Ludwigson, Ludwik, Swift, and Voce models [28]. Ji et al. analyzed the damage evolution and fracture behavior of 7075 aluminum alloy sheets during the tensile process under different temperatures through the GTN damage model [29]. Zhu et al. explained the stress flow behavior of stainless steel in the extrusion forming process of spherical plain bearings at room temperature using the Ludwik, Swift, H-S, and Voce models [30]. In addition, considering the partially melted zone properties, stress concentration at the weld toe, dimensions of the tensile test specimen, and welding residual stress, Wan and Wang et al. researched the tensile properties and fracture behavior of 2219-T8 aluminum alloy TIG welding joints with the J-C criterion and analyzed the constitutive behavior and damage mechanism of the eutectic structure and Al matrix [5,31,32]. As to the abovementioned literature, the constitutive model and damage model have critical roles in the depiction of the flow stress and fracture behaviors of metal material sheets. However, from the above reviews, the tensile properties and hardening behaviors of welded joints, which were affected by different welding currents under welding deformation of thin-walled part structures, were less focused.

Therefore, according to the abovementioned problems and the emphasis on research, there are some works in this paper as follows: (1) A finite element model is established to investigate the welding deformation conditions of T-joint thin-walled parts; on this basis, the tensile test simulation of T-welded joint specimens is carried out. (2) In combination with numerical simulation and tensile experiment, the tensile property variation and fracture behaviors of TIG T-welded joints for tensile specimens for different welding currents are investigated, employing different fracture criteria that are formed by multiply strain hardening models and constitutive models, combining with a fracture failure model. (3) The different fracture criteria are evaluated and compared to verify their accuracy and effectiveness under different welding currents. On the whole, it will provide an analytical foundation for better understanding the tensile properties and fracture behaviors of TIG T-welded joints under different welding currents and further improving the welding assembly quality of thin-walled parts in the future.

## 2. Experiment Procedure

In this work, the welding deformation and mechanical properties of a T-joint thin-walled part from an antenna structure were analyzed. AA6061-T6 of a thin-walled part with a thickness of 2 mm was selected as the experimental material, which was welded by welding wire ER4043 with a diameter of 2.0 mm. It is a series of Al-Si welding wires that contain about 4.5–6.0% Si, which can provide a large number of low melting point eutectic materials with good fluidity in the liquid state and smaller shrinkage during solidification in the welding process. Hereby, the welding wire ER4043 was adopted, combining its advantages with this work.

Before the actual welding procedure for T-joint thin-walled parts, the two top zones of two sides for the web plate and base plate were fixed through spot welding to facilitate welding assembly. They were in the red zone, as shown in Figure 1a. Subsequently, the web plate and base plate were welded by manual TIG procedure. The actual welding currents were set at 100 A, 140 A, 170 A, and 220 A, and the Argon gas flow was 10–12 L/min. The web plate and base plate of the thin-walled T-joint part were welded together, as shown in Figure 1b, to obtain the tensile specimens. In addition, the structure size of the T-joint thin-walled part and welding direction with a blue arrow are shown in Figure 1c.

Regarding the preparation of the tensile specimens, tensile test specimens were prepared perpendicular to the weld seam. Moreover, every T-joint thin-walled part structure was evenly cut into three specimens with a wire-cutting machine tool, as shown in Figure 2. What is more, the cross-section area of the specimens was 18 mm × 2 mm, the parallel segment length was 30 mm, and the gauge length of the test specimens was 25 mm using an electronic extensometer (its specification and model is YYU-12.5/25). Subsequently, the tensile test specimens were clamped with fixtures on the SANS electronic universal testing machine, and their tensile rate was 1 mm/min, as shown in Figure 3. On this basis, force-displacement data under different welding currents were acquired, whether through the tensile test of T-welded joint specimens or the subsequent numerical simulation. According to these data, the true stress-true strain value is calculated using the following equations [33]:(1)$$\left\{\begin{array}{c} \sigma_{\mathrm{true}}=\sigma \left(1+\varepsilon \right)\\ \varepsilon_{\mathrm{true}}=\ln\left(1+\varepsilon \right) \end{array}\right.$$
where σ and ε are the engineering stress and engineering strain values, respectively.

## 3. Numerical Simulation of T-Joint Thin-Walled Part Welding

In order to analyze the tensile properties of the T-welded joint under welding deformation, the welding deformation of the T-joint thin-walled part structure, which was affected by welding current or heat input, was investigated by FE simulation. When the welding deformation of T-joint thin-walled parts was analyzed using FE simulation, the spot welding zone of the two sides of the thin-walled parts was the fixed constraint condition, as shown in Figure 1a. When the FE simulation was carried out, the welding voltage of 20 V, the welding speed of about 3 mm/s, and the welding efficiency of 0.75 were accepted. The corresponding welding currents were matched with welding experiments on thin-walled parts. What is more, the front length of 1.5 mm, rear length of 6 mm, width of 2.5 mm, and depth of 2.5 mm for the heat source model of the used double ellipsoid were adopted in previous work [1,2]. What is more, the total welding time was 70 s, and the cooling time was 200 s.

Additionally, when FE simulation was carried out, the double ellipsoidal heat source model was adopted, and the power density distribution function of the front and rear half ellipsoids is as follows [1,34]:(2)$$q\left(f\right)=\frac{6\sqrt{3}f_{f}Q_{f}}{\pi^{3/2}a_{f}bc}\exp\left(-3\left({\left(\frac{x}{a_{f}}\right)}^{2}+{\left(\frac{y}{b}\right)}^{2}+{\left(\frac{z}{c}\right)}^{2}\right)\right)$$
(3)$$q\left(r\right)=\frac{6\sqrt{3}f_{r}Q_{r}}{\pi^{3/2}a_{r}bc}\exp\left(-3\left({\left(\frac{x}{a_{r}}\right)}^{2}+{\left(\frac{y}{b}\right)}^{2}+{\left(\frac{z}{c}\right)}^{2}\right)\right)$$
where *Q_f_* and *Q_r_* are the heat inputs of the front and rear parts, and the calculation of the total heat input *Q* is as follows [35,36]:(4)Q=ζ⋅U⋅I

What is more, *af* and *ar* are the lengths of the front and rear parts, and *b* and *c* are the width and depth, respectively. The energy fractions of the front and rear half ellipsoids were *ff* and *fr*, and *ff* + *fr* = 2. U, I, and ζ are welding voltage, welding current, and welding efficiency, respectively. Therefore, FE simulation was implemented to obtain the welding properties of a T-joint thin-walled part with the built heat source model through the DFLUX subroutine employing FORTRAN combined with ABAQUS.

Furthermore, the thermal and mechanical properties of AA6061-T6 with temperature were essential for FE simulation using ABAQUS. According to the parameters in the literature [37], the thermal and mechanical property variation curves depending on the temperature are shown in Figure 4, and the other relative parameter settings are shown in Table 1.

### 3.1. Tensile Test Simulation

In this work, in order to accurately simulate the tensile behavior of a T-welded joint at room temperature, the tensile test specimen model was also cut from the T-joint thin-walled part structure model of the welding-deformed joint. When the tensile test simulation using dynamic explicit analysis of finite elements was conducted, the left zone of the tensile test specimen model was fixed, and the tensile velocity for the other side was applied, acting on the tensile region as shown in Figure 5.

Simultaneously, in order to save calculation time and ensure simulation accuracy, when the tensile testing simulation was carried out, the tensile velocity was substantially increased compared to the actual tensile test velocity. At this time, the internal energy ratio of the kinetic energy of the tensile specimens remained below 5% through ABAQUS/Explicit analysis. After all the tensile specimens’ simulation was completed, the simulation time was multiplied by the conversion coefficient (the velocity ratio of simulation and tensile test), which was the conversion time. On this foundation, in this work, it was assumed that under the same conversion time, the tensile displacement was the same and the deformation state of the tensile specimen was also the same; moreover, the yield stress and tensile strength remained unchanged, respectively. Hereby, when the relationship between stress and strain was handled, the obtained displacement value for different hardening models through FE simulation was selected, which was the same as the displacement value measured in the tensile test at the same time.

Due to the obvious irregularity of the welding-deformed model of the T-joint thin-walled part, when the tensile test specimen models were cut from the welding structure of the T-joint thin-walled part, the C3D8R hexahedron element combined with the C3D4 tetrahedron element were used to complete the mesh division of the finite element. The FE model of the tensile test specimen was shown in Figure 5, and the number of mesh elements in the tensile test specimen from the welding-deformed structure model of the T-joint thin-walled part that was obtained under different welding currents was shown in Table 2. In this work, it was assumed that the numerical simulation results were not affected by the number of mesh elements.

Hereby, the tensile properties and working hardening behaviors of a T-welded joint were investigated using several different fracture criteria, namely the J-C model, Swift model, Voce model, and H-S model, and their linear combined model, respectively.

### 3.2. The Constitutive Model and Failure Model

#### 3.2.1. The Johnson-Cook Model

In this work, the employed constitutive model was simplified to go deeply into the tensile behaviors of T-welded joints for thin-walled parts. The initial model was proposed by Johnson and Cook [38]. Its mathematical relationship was as follows:(5)$$\sigma =\left(A+B{\left({\bar{\varepsilon }}^{p}\right)}^{n}\right)\left(1+C\ln\frac{\dot{\varepsilon }}{{\dot{\varepsilon }}_{0}}\right){\left(1-\frac{T-T_{r}}{T_{m}-T_{r}}\right)}^{m}$$
where σ is the equivalent stress, ${\bar{\varepsilon }}^{p}$ is the equivalent plastic strain, $\dot{\varepsilon }$ is the plastic strain rate, and ${\dot{\varepsilon }}_{0}$ is the reference strain rate. T is the current temperature, $T_{m}$ is the material melting temperature, and $T_{r}$ is room temperature. The model parameters A, B, C, m, and n could be obtained by the least-squares method using the stress-strain data from tensile tests or simulations of T-welded joint specimens.

In this work, the thermal softening effect was not considered through FE simulation, and the tensile test of T-welded joint specimens was carried out at room temperature. Therefore, the model of Equation (5) could be simplified to a relationship formula related to the strain and strain rate, as shown in Equation (6). What is more, the different plastic strain rates are not taken into consideration in this work; therefore, the original J-C equation is evolved into Equation (7).
(6)$$\sigma =\left(A+B{\left({\bar{\varepsilon }}^{p}\right)}^{n}\right)\left(1+C\ln\frac{\dot{\varepsilon }}{{\dot{\varepsilon }}_{0}}\right)$$
(7)$$\sigma =A+B{\left({\bar{\varepsilon }}^{p}\right)}^{n}$$

#### 3.2.2. The Damage and Fracture Failure Model

Simultaneously, Johnson and Cook proposed plasticity and failure models considering multiple factors; the critical equivalent fracture strain was taken as the damage characterization variable. Its expression is as follows [5,38]:(8)$${\bar{\varepsilon }}_{f}=\left[D_{1}+D_{2}e^{D_{3}\eta }\right]\left[1+D_{4}\ln\left(\frac{\dot{\varepsilon }}{{\dot{\varepsilon }}_{0}}\right)\right]\left[1+D_{5}\cdot \left(\frac{T-T_{r}}{T_{m}-T_{r}}\right)\right]$$
where $D_{1}$, $D_{2}$, $D_{3}$, $D_{4}$, and $D_{5}$ are failure parameters, η is the stress triaxiality (ratio of hydrostatic stress to equivalent von Mises stress), which could be obtained through the combination of numerical simulation and experiment data in this work.
(9)$$\eta =\frac{\sigma_{1}+\sigma_{2}+\sigma_{3}}{3\sqrt{\left[{\left(\sigma_{1}-\sigma_{2}\right)}^{2}+{\left(\sigma_{2}-\sigma_{3}\right)}^{2}+{\left(\sigma_{3}-\sigma_{1}\right)}^{2}\right]/2}}$$
where $\sigma_{1}$, $\sigma_{2}$ and $\sigma_{3}$ are the principle stresses.

What is more, the damage value D is an accumulation value. When D=1, the metal part will be occurring to fracture failure. The damage formula is as follows [39]:(10)$$D=\sum \frac{\Delta {\bar{\varepsilon }}^{p}}{{\bar{\varepsilon }}^{f}}$$
where $\Delta {\bar{\varepsilon }}^{p}$ is the equivalent plastic strain increment of an integral cycle.

According to the above models, it was clearly revealed that the equivalent fracture strain was determined by stain rate and temperature. Due to the constant strain rate and invariant experiment temperature in this work, only the effect of stress triaxiality was taken into consideration. Its expression is as follows:(11)$${\bar{\varepsilon }}_{f}=D_{1}+D_{2}e^{D_{3}\eta }$$

#### 3.2.3. The Swift, Voce, and Hockett-Sherby Model

In this work, the Swift model [40], Voce model [41], and Hockett-Sherby (H-S) model [42], which described the flow stress of metal material, were also employed to describe the strain behaviors at the strain-hardening stage of the tensile properties’ curves of T-welded joint specimens. Their formulas were given as follows:(12)$$\sigma_{\mathrm{Swift}}=\phi \cdot {\left(\varepsilon_{s}+{\bar{\varepsilon }}^{p}\right)}^{n}$$
(13)$$\sigma_{\mathrm{Voce}}=\sigma_{s}+\psi \cdot \left(1-e^{-m\cdot {\bar{\varepsilon }}^{p}}\right)$$
(14)$$\sigma_{H-S}=\sigma_{s}+\xi \cdot \left(1-e^{-k\cdot {\bar{\varepsilon }}_{p}^{n}}\right)$$
where $\sigma_{s}$ is the yield stress, $\varepsilon_{s}$ is yield strain, ϕ,ψ, and ξ are fitting parameters, and m,n, and k are the hardening factor constants, respectively.

#### 3.2.4. The Novel Combination Hardening Models

It was necessary to modify the original J-C model to characterize the stress-strain relationship accurately as well as conduct precise FE simulations. In this work, a novel linear combination of the Swift-Voce hardening models and the H-S-Voce and Swift-H-S hardening models was employed to describe the strain hardening behaviors and to fit the true stress and plastic strain data, which combined the tension test with the simulation force-displacement curve of a T-welded joint. The constructed working hardening equations were as indicated in Equations (15)–(17).
(15)$$\begin{array}{l} \sigma \left({\bar{\varepsilon }}^{p}\right)=\lambda \sigma_{\mathrm{swift}}\left({\bar{\varepsilon }}^{p}\right)+\left(1-\lambda \right)\sigma_{\mathrm{Voce}}\left({\bar{\varepsilon }}^{p}\right)=\\ \lambda \left[\phi {\left(\varepsilon_{s}+{\bar{\varepsilon }}^{p}\right)}^{n}\right]+\left(1-\lambda \right)\left[\sigma_{s}+\psi \left(1-e^{-m\cdot {\bar{\varepsilon }}^{p}}\right)\right]\\ =\lambda \phi {\left(\varepsilon_{s}+{\bar{\varepsilon }}^{p}\right)}^{n}-\left(1-\lambda \right)\psi e^{-m\cdot {\bar{\varepsilon }}^{p}}+\left(1-\lambda \right)\left(\sigma_{s}+\psi \right) \end{array}$$
(16)$$\begin{array}{l} \sigma \left({\bar{\varepsilon }}^{p}\right)=\lambda \sigma_{H-S}\left({\bar{\varepsilon }}^{p}\right)+\left(1-\lambda \right)\sigma_{Voce}\left({\bar{\varepsilon }}^{p}\right)=\\ \lambda \left[\sigma_{s}+\xi \left(1-e^{-k\cdot {\bar{\varepsilon }}_{p}^{n}}\right)\right]+\left(1-\lambda \right)\left[\sigma_{s}+\psi \left(1-e^{-m\cdot {\bar{\varepsilon }}^{p}}\right)\right]\\ =-\lambda \xi e^{-k\cdot {\bar{\varepsilon }}_{p}^{n}}-\left(1-\lambda \right)\psi e^{-m\cdot {\bar{\varepsilon }}^{p}}+\lambda \left(\xi -\psi \right)+\sigma_{s}+\psi \end{array}$$
(17)$$\begin{array}{l} \sigma \left({\bar{\varepsilon }}^{p}\right)=\lambda \sigma_{\mathrm{Swift}}\left({\bar{\varepsilon }}^{p}\right)+\left(1-\lambda \right)\sigma_{H-S}\left({\bar{\varepsilon }}^{p}\right)=\\ \lambda \left[\phi {\left(\varepsilon_{s}+{\bar{\varepsilon }}^{p}\right)}^{n_{1}}\right]+\left(1-\lambda \right)\left[\sigma_{s}+\xi \left(1-e^{-k\cdot {\bar{\varepsilon }}_{p}^{n_{2}}}\right)\right]\\ =\lambda \phi {\left(\varepsilon_{s}+{\bar{\varepsilon }}^{p}\right)}^{n_{1}}-\left(1-\lambda \right)\xi e^{-k\cdot {\bar{\varepsilon }}_{p}^{n_{2}}}+\left(1-\lambda \right)\left(\sigma_{s}+\xi \right) \end{array}$$
where the coefficient λ is a weighting factor between 0 and 1. In this work, the coefficient λ was selected as 0.2, 0.4, 0.6, and 0.8 to establish the above hardening equations for analyzing the tensile behaviors of T-joint welded specimens.

According to the abovementioned equations of the constitutive model and failure model, the tensile testing simulation was conducted by coding the VUMAT and VUSDFLD subroutines using ABAQUS.

In addition, the accuracy of the abovementioned model was judged and validated by the correlation coefficient and mean squared error (MSE) in this work, as shown in Equations (18) and (19).
(18)$$R=\frac{\sum_{i=1}^{N}\left(\sigma_{\exp}^{i}-{\bar{\sigma }}_{\mathrm{pre}}\right)\left(\sigma_{\mathrm{pre}}^{i}-{\bar{\sigma }}_{\mathrm{pre}}\right)}{\sqrt{\sum_{i=1}^{N}{\left(\sigma_{\exp}^{i}-{\bar{\sigma }}_{\exp}\right)}^{2}}\cdot \sqrt{\sum_{i=1}^{N}{\left(\sigma_{\mathrm{pre}}^{i}-{\bar{\sigma }}_{\mathrm{pre}}\right)}^{2}}}$$
(19)$$MSE=\frac{1}{N}{\sum_{i=1}^{N}\left(\frac{\sigma_{\exp}^{i}-\sigma_{\mathrm{pre}}^{i}}{\sigma_{\mathrm{pre}}^{i}}\right)}^{2}$$
where *N* is the number of data points, $\sigma_{\exp}$ and ${\bar{\sigma }}_{\exp}$ are respectively the true stress and mean stress through the tensile experiment, $\sigma_{\mathrm{pre}}$ and ${\bar{\sigma }}_{\mathrm{pre}}$ are respectively the true stress and mean true stress obtained by the above constitutive model.

## 4. Results and Discussion

According to the abovementioned welding experiment of T-joint thin-walled parts, the tensile test of T-welded joint specimens, their numerical simulation conditions, multiply hardening models, failure models, the results of welding deformation, the stress-strain relationship of T-welded joint specimens, etc. were obtained to investigate the tensile properties and fracture behaviors that were affected by different fracture criteria.

### 4.1. Welding Deformation

Due to the effect of welding deformation on the T-joint thin-walled part structure under different welding currents, the obtained tensile specimen models were also deformed. In order to analyze the deformation of a T-welded joint, the node deformation of seven different paths on the base plate of the tensile specimen model was investigated; the first and second paths were, respectively, 6 mm and 10 mm from the left side of the web plate, and the third and fourth paths were also so. The sixth path was at the middle of the base plate, and the fifth and seventh paths were located at the edge of the base plate after cutting the T-joint thin-walled part, as shown in Figure 6. According to the above welding process simulation, the deformed tensile specimen models were obtained by cutting the T-joint thin-walled part structure model after welding deformation. On this basis, the deformation variation curves for different paths on the tensile specimen model are shown in Figure 7.

For the tensile specimens of T-welded joints, Figure 7 illustrated that the T-joint thin-walled parts of 2 mm thickness had a larger deformation under different welding currents. On the whole, the welding deformation or warp degree of the T-joint thin-walled parts was increasing with the increment of welding current to a certain extent. Specifically, in Figure 7a,d, when the welding current was 100 A, the mean values of node deformation on the first and fourth paths were about 1.170 mm and 1.089 mm, respectively, and the mean values of node deformation on corresponding paths 2 and 3 were about 1.264 mm and 1.209 mm, respectively. However, compared with 100 A, when the welding current was 140 A, their deformation on corresponding paths 1, 2, 3, and 4 was respectively more than 13%, 48%, 44%, and 53%, as shown in Figure 7a–d. When the welding current was 170 A or 220 A, their deformation was almost larger than 2 mm, even if most of the node deformation on the paths 5, 6, and 7 in Figure 7e–g was also greater than 2 mm, and the maximum deformation values were respectively about 2.24 mm and 2.7 mm for 170 A and 220 A. These results showed that the welding deformation was greatly affected by the welding current, which was caused by the increment of welding heat input, and when welding cooled, the weld metal began to solidify and shrink, resulting in the warpage and distortion of the aluminum alloy thin-walled part. Given that, the mechanical properties and tensile behaviors of the T-welded joint specimen were conducted and investigated through the combination of numerical simulation and experiment under the effect of welding deformation in this work.

### 4.2. Determination of Constitutive Equations, Hardening Equations, and Failure Models

In order to accurately characterize the stress-strain relationship according to the tensile test experiment data of T-welded joint specimens, the modified J-C constitutive equations under different welding currents were built through the obtained fitting parameters, as shown in Equation (20). Furthermore, Equation (20) was applied in writing a subroutine for FE simulation using ABAQUS, which aimed at tensile test simulation to obtain the stress-strain relationship and analyze the tensile properties of T-welded joint specimens. Simultaneously, according to the experiment results of tensile test specimens, the true stress-true strain curves under different welding currents were shown in Figure 8a, and the true stress-plastic strain curves in the plastic deformation stage were shown in Figure 8b.
(20)$$\left\{\begin{array}{c} \sigma_{J-C-100}=425.302{\left({\bar{\varepsilon }}^{p}\right)}^{0.69793}+139.5\\ \sigma_{J-C-140}=399.256{\left({\bar{\varepsilon }}^{p}\right)}^{0.54086}+100.5\\ \sigma_{J-C-170}=396.115{\left({\bar{\varepsilon }}^{p}\right)}^{0.51363}+97.5\\ \sigma_{J-C-220}=382.812{\left({\bar{\varepsilon }}^{p}\right)}^{0.54037}+112.5 \end{array}\right.$$

Figure 8a shows the relationship between true stress and true train for different welding currents through a tensile test. Likewise, the stress and strain were also subjected to the effect of welding current, especially the variety of fracture strain positions that emerged. Moreover, those curves were all composed of the linearity stage, the strain-hardening stage, and the damage failure stage. What is more, according to the linear fitting of stress-strain curves and the 0.2% offset method, the yield stress for different welding currents was obtained. They were about 139.5 MPa, 100.5 MPa, 97.5 MPa, and 112.5 MPa, corresponding to 100 A, 140 A, 170 A, and 220 A. In addition, the obtained tensile strength was about 239.2 MPa, 242.3 MPa, 233.8 MPa, and 235.5 MPa, which accounts for 77.2%, 78.2%, 75.4%, and 76.0% of the base metal tensile strength of the thin-walled part.

Figure 8b revealed that in the strain-hardening stage in Figure 8a, the true stress increased with the increment of plastic strain of T-welded joint specimens through a modified and simplified J-C model and tensile test at room temperature. It was also shown that the results of stress-plastic strain simulated with the MJ-C model were not much different from the tensile specimen’s test results under the influence of respective welding currents, and the variation trend of the simulated curves was well in agreement with the experiment curve. It was also further illustrated that the MJ-C model could also accurately describe the strain hardening behaviors of T-welded joints with FE simulation.

#### 4.2.1. The Attainment of Hardening Equations’ Parameters under Different Welding Currents

According to the true stress-strain curves in Figure 8a, the yield stress and yield strain were determined, and the nonlinear fitting of plastic strain-stress data was conducted by the built-in Levenberg-Marquardt optimization algorithm using Origin to obtain the relative parameters in hardening equations. The obtained parameters of the Swift model, the Voce model, and the H-S model are respectively shown in Appendix A. It is noted that these parameters can be seen in the attached Appendix A, including Appendix A. Hereby, the equations of Swift, Voce model, and H-S were established and determined to apply for analyzing the tensile behaviors of T-welded joints through FE simulation.

As such, when the welding current is 100 A, 140 A, 170 A, and 220 A, the obtained yield stress is, $\sigma_{s}=139.5\mathrm{MPa},100.5\mathrm{MPa},97.5\mathrm{MPa}$ and 112.5MPa, respectively, through tensile test of T-welded joint specimens. When the coefficient parameter value was λ= 0.2 in Equation (15), the obtained Swift-Voce combination equation was shown in Equation (21). According to the coefficient parameter values of λ= 0.4, 0.6, and 0.8, and according to the relative parameters in Appendix A, the corresponding Swift-Voce combination model was also constructed to simulate the tensile test of a T-welded joint.
(21)$$\left\{\begin{array}{c} \sigma \left({\bar{\varepsilon }}^{p}\right)\left|{}_{100}^{0.2S-V}\right.=49.761\left[{\left(0.00396+{\bar{\varepsilon }}^{p}\right)}^{0.11754}\right]-121.9618e^{-9.6104{\bar{\varepsilon }}^{p}}+233.5618\\ \sigma \left({\bar{\varepsilon }}^{p}\right)\left|{}_{140}^{0.2S-V}\right.=232.046\left[{\left(0.00434+{\bar{\varepsilon }}^{p}\right)}^{0.45896}\right]-61.8332e^{-21.47453{\bar{\varepsilon }}^{p}}+142.2332\\ \sigma \left({\bar{\varepsilon }}^{p}\right)\left|{}_{170}^{0.2S-V}\right.=310.1848\left[{\left(0.0042+{\bar{\varepsilon }}^{p}\right)}^{0.52708}\right]-46.5126e^{-43.84332{\bar{\varepsilon }}^{p}}+124.5126\\ \sigma \left({\bar{\varepsilon }}^{p}\right)\left|{}_{220}^{0.2S-V}\right.=240.1872\left[{\left(0.00446+{\bar{\varepsilon }}^{p}\right)}^{0.46567}\right]-49.9332e^{-32.27892{\bar{\varepsilon }}^{p}}+139.9332 \end{array}\right.$$

Likewise, when the coefficient parameter value was λ= 0.2, the Voce-H-S combination model was established, as shown in Equation (22). For the coefficient parameter values λ= 0.4, 0.6, and 0.8, according to the relative parameters in Appendix A, the corresponding Voce-H-S combination model was also determined.
(22)$$\left\{\begin{array}{c} \sigma \left({\bar{\varepsilon }}^{p}\right)\left|{}_{100}^{0.2\mathrm{HS}-V}\right.=-111.115e^{-16.22961{\bar{\varepsilon }}_{p}{}^{1.18918}}-13.46275e^{-61.84569{\bar{\varepsilon }}^{p}}+264.0777\\ \sigma \left({\bar{\varepsilon }}^{p}\right)\left|{}_{140}^{0.2\mathrm{HS}-V}\right.=-12.48452e^{-301412.4935{\bar{\varepsilon }}_{p}{}^{2.67975}}-146.795e^{-12.52134{\bar{\varepsilon }}^{p}}+259.7795\\ \sigma \left({\bar{\varepsilon }}^{p}\right)\left|{}_{170}^{0.2\mathrm{HS}-V}\right.=-18.59127e^{-153626.2471{\bar{\varepsilon }}_{p}{}^{2.50682}}-135.0142e^{-14.33071{\bar{\varepsilon }}^{p}}+251.1054\\ \sigma \left({\bar{\varepsilon }}^{p}\right)\left|{}_{220}^{0.2\mathrm{HS}-V}\right.=-11.60589e^{-62215.92509{\bar{\varepsilon }}_{p}{}^{2.37628}}-124.6234e^{-15.50425{\bar{\varepsilon }}^{p}}+248.7293 \end{array}\right.$$

In addition, for the Swift-H-S combination equation in Equation (17), the yield strain $\varepsilon_{s}$ was still 0.00396, 0.00434, 0.0042, and 0.00446. When the coefficient parameter value was λ= 0.2, the Swift-H-S combination model was established, as shown in Equation (23). For the coefficient parameter values λ= 0.4, 0.6, and 0.8, the parameters for the corresponding equation were shown in Appendix A.
(23)$$\left\{\begin{array}{c} \sigma \left({\bar{\varepsilon }}^{p}\right)\left|{}_{100}^{0.2S-\mathrm{HS}}\right.=262.9524{\left(0.00396+{\bar{\varepsilon }}^{p}\right)}^{0.43378}-16.42514e^{-252.005{\bar{\varepsilon }}_{p}^{2.00861}}+128.0251\\ \sigma \left({\bar{\varepsilon }}^{p}\right)\left|{}_{140}^{0.2S-\mathrm{HS}}\right.=10.6464{\left(0.00434+{\bar{\varepsilon }}^{p}\right)}^{0.02431}-200.2726e^{-5.15094{\bar{\varepsilon }}_{p}^{0.71112}}+280.6726\\ \sigma \left({\bar{\varepsilon }}^{p}\right)\left|{}_{170}^{0.2S-\mathrm{HS}}\right.=2.23\times 10^{-7}{\left(0.0042+{\bar{\varepsilon }}^{p}\right)}^{23.99458}-223.3491e^{-3.85751{\bar{\varepsilon }}_{p}^{0.59019}}+301.3491\\ \sigma \left({\bar{\varepsilon }}^{p}\right)\left|{}_{220}^{0.2S-\mathrm{HS}}\right.=5.0893{\left(0.00396+{\bar{\varepsilon }}^{p}\right)}^{-0.06905}-181.8256e^{-5.49884{\bar{\varepsilon }}_{p}^{0.66766}}+271.8256 \end{array}\right.$$

According to the corresponding model parameter values for different welding currents in the abovementioned tables, they were substituted into Equations (15)–(17), and all combination equations were built as working hardening equations to investigate the tensile behaviors of T-welded joints combined with the fracture failure model by the ABAQUS subroutine.

#### 4.2.2. Fracture Failure Model

According to Equation (11), the fracture strain value and the stress triaxiality of tensile specimens at fracture time combined with tensile test and FE simulation were extracted to fit and obtain the failure parameter values of $D_{1}$, $D_{2}$, and $D_{3}$.

The fitted curve between the stress trixiality and fracture strain is shown in Figure 9. According to the fitted curve, the obtained parameter values of the fracture model were $D_{1}=0.10865$, $D_{2}=0.93018$, and $D_{3}=$ −7.8972, respectively. Therefore, the obtained relationship equation between fracture strain and stress triaxiality was as follows:
(24)$${\bar{\varepsilon }}_{f}=0.13363+0.20588e^{-7.8972\eta }$$

Hereby, the abovementioned hardening equations, combined with the fracture failure model, were conducted by numerical simulation. Simultaneously, it should be noted that due to the complex stress states of the T-welded joint and the great effect of stress triaxiality on the plastic strain in the strain-hardening stage before fracture failure, it could effectively report the influence of different stress states on the failure strain of the T-welded joint, considering the stress triaxiality. Moreover, because the stress triaxiality was the ratio of hydrostatic stress to equivalent von Mises stress, the equivalent von Mises stress was essential for investigating the fracture behaviors and mechanical properties of the T-welded joint specimens. Therefore, in this section, taking the Misses stress before tensile fracture failure of T-welded joint as an example, when welding current was 220 A and parameter value λ = 0.2 using different hardening models, its nephogram was shown in Figure 10. It could be seen that the variation distribution of von Mises stress was observed, and the von Mises stress data was also extracted from the nephogram to analyze the stress triaxiality before fracture failure. The von Mises stress distribution, which was obtained through other welding currents, and the parameter coefficient value λ were similar.

Figure 10 proclaimed that under the same welding current, there was some discrepancy between the von Mises stress achieved with the J-C model, Swift model, H-S model, Voce model, and their combination model. From the perspective of the maximum von Mises stress value, they were respectively 238.8 MPa, 236.5 MPa, 232.9 MPa, 230.9 MPa, 233 MPa, 234.7 MPa, 234.2 MPa, 234.1 MPa, 234.2 MPa, and 234.3 MPa. For the single model in Figure 10a–d, comparing with the result of the simulation with the J-C model, the maximum relative error was not more than 4%. Moreover, for the combination model in Figure 10e–j, the effect was found to be small. For the same combination model, the influence of different parameter values (λ) on the simulation results of von Mises stress before fracture failure of a T-welded joint was very tiny. It was shown that those models were very suitable for depicting the plastic strain in the strain-hardening stage before fracture failure of the T-welded joint. These results were also greatly essential for obtaining stress triaxiality and investigating the tensile behaviors of T-welded joints.

### 4.3. Effect of Different Welding Currents and Fracture Criteria on Tensile Behaviors

The abovementioned hardening models and fracture failure models were implemented into ABAQUS, and then the FE simulation was conducted with the VUMAT and VUSDFLD subroutines to obtain the load-displacement data of tensile specimens for a T-welded joint. They were calculated with Equation (1) to achieve the true stress-true strain of the simulation with different models and tensile tests at different welding currents, as shown in Figure 11 and Figure 12. Based on this, the tensile property results are shown in Figure 13.

Figure 11 indicated that the true stress and strain were really affected by the single model or combination model through the variation relationship between true stress and true strain simulated with different models, especially those true stress-trues strain trends of the linearity stage and strain-hardening stage that were greatly similar and closer to the tensile test results. In terms of the initiated fracture strain, there were some discrepancies compared with tensile test results to a certain extent, but the differences were not much larger under the aforementioned assumptions. Even for the same combination model, the initiated fracture strain also had a tiny discrepancy when it had different coefficient values (λ) (Figure 11b,d,f,h). However, under the same welding current and combination model, comparing with the parameter value λ = 0.2, the relative error of the initiated fracture strain with the H-S-Voce model and Swift-H-S model when welding current was 100 A was only about 1%, and for the Swift-Voce model, this maximum error was also about 5%. When welding currents of 140 A, 170 A, and 220 A were used, the maximum relative error value of the initiated fracture strain was also not beyond 5%, compared with the results corresponding to the coefficient value λ = 0.2. It was also confirmed that the stress and strain results obtained by those models were relatively matched with the tensile test results to a certain degree, and the different fracture criteria formed by combining hardening models and failure models were also greatly suitable for the tensile behavior simulation of a welded joint of a thin-walled joint.

Simultaneously, in order to further compare and investigate the effect of different welding currents on the stress and strain under the same fracture criteria and different welding currents, the partial true stress-true strain curves were drawn, as shown in Figure 12. The overall variation trend was greatly similar, but the stress-strain had some differences at the strain-hardening stage and fracture failure. Specifically, when the single model was employed as shown in Figure 12a–d, it could be seen that there was some discrepancy in fracture strain for different welding currents, and the initial fracture strain corresponding to welding currents of 100 A, 140 A, and 170 A decreased when welding current increased compared to the results of welding current of 220 A. Based on the initial fracture strain corresponding to a welding current of 220 A, for welding currents of 100 A, 140 A, and 170 A, for the adopted J-C model, the initial fracture strain increased by about 31.3%, 18.9%, and 26%, respectively. The largest increment rate was 43.2%, which occurred at a welding current of 100 A, employing the H-S model in Figure 12b. For the adopted Swift and Voce single models, the increment rate was also variable within the range of 16–35%. Likewise, for welding currents of 100 A, 140 A, and 170 A for the adopted H-S-Voce combination model and the same parameter coefficient value in Figure 12e,f, the maximum increment rate of initial fracture strain was about 31%, and the minimum proportion was also more than 15%. Other results simulated with other single models and combination models under different coefficient values (λ) were similar. It was clearly shown that the variation of stress-strain and initial fracture strain were deeply affected by welding current, whether employing the single model or the combination model. These results will also provide an analysis for the optimization of welding process parameters to investigate the tensile behavior of welded joints for thin-walled parts in the future.

Figure 13 illustrates that tensile properties were affected by the welding current, whether the simulated results were with a single model or a combination model and tensile test results. What is more, under the abovementioned assumptions, for yield stress of T-welded joint specimens in Figure 13a,b, when the welding current was 100A, comparing with the results of tensile tests of T-welded joint specimens, the relative error values were about 12.6%, 10.1%, 4.0%, and 9.9% simulated with the J-C, H-S, Swift, and Voce models. The mean values of relative errors were 10%, 9.6%, and 9.7% with the H-S-Voce, Swift-H-S, and Swift-Voce combination models when coefficient values λ = 0.2, 0.4, 0.6, and 0.8, respectively. When the welding current was 140 A, except for the relative error values of 13.1% and 19.5% for J-C and H-S models, the other relative error values were all about less than 10% when compared with the tensile test results, whether single models or combination models. When the welding current was 170 A, except for the relative error value of 13.4% for the J-C model, including the welding current of 220 A, the other relative error values were all about 5%. In addition, when the welding currents were 170 A and 220 A, the minimum relative error values were about 2.3% and 0.8%, which were simulated with the J-C single model and Swift-Voce combination model when parameter value λ = 0.8. Taken as a whole, it was shown that the combination models were in better agreement with the experiment value than the single model, but the simulation results were within the allowable range with the single model.

For the tensile strength of T-welded joint specimens in Figure 13c,d, compared with the experiment results, the difference between the simulation results of all models was not more than 18 MPa, and there was a minor discrepancy between all models, especially the minimal differences among the results of simulation with different combination models. In Figure 13c, when the welding current was 100 A, the relative error value was 7.3% for the J-C model and H-S model, which compared with the experiment results, was below 3% under other welding currents. However, In Figure 13d, the mean value of relative error was about 5.6%, 5.83%, and 5.85% under a welding current of 100 A with the H-S-Voce, Swift-H-S, and Swift-Voce combination models when the coefficient value λ = 0.2, 0.4, 0.6, and 0.8. When the welding current was 140 A, their mean errors were 1.94%, 5.6%, 2%, and 1.98%. When the welding current was 170 A, their mean errors were about 3%. When the welding current was 220 A, their mean errors were only below 1% with all combination models, which was greatly closer to the tensile strength value of the tensile experiment. From this analysis, it could be seen that the larger the welding current, the lesser the simulated error value with different fracture criteria, which were formed by combining hardening models and failure models. No greatly apparent influence of parameter value λ on the simulation value was further confirmed under the same fracture criteria. Moreover, it was also further proven that the results obtained by the combined models were relatively more in accord with the tensile test results than those achieved by different single models.

### 4.4. Evaluation of Different Fracture Criterions on the Prediction of Tensile Properties

In order to further investigate the prediction accuracy of different fracture criteria, according to Equations (18) and (19), the correlation coefficient value and MSE value were calculated as shown in Appendix A. Additionally, it was known that the closer the correlation coefficient value is to 1, the closer the relationship between the prediction value of models and the experiment value, the more accurate the model. When the MSE value was smaller, the prediction accuracy of the model was higher. From this point of view, the obtained results simulated with multiplied single models and combination models were evaluated. In Appendix A, it could be found that there was less difference in the achieved correlation coefficient value simulated with different fracture criteria under different welding currents. The maximum correlation coefficient values (R) were 0.997688 and 0.99771 with the Swift-Voce and Swift-H-S combination models under welding currents of 100 A, 140 A, and λ = 0.2. The maximum correlation coefficient value R was 0.995262 with the Swift-Voce model for welding current of 170 A and λ = 0.8. However, beyond that, when λ = 0.6 and 0.8, the obtained maximum correlation coefficient value was almost the same with the Swift-H-S combination model, all of which were about 0.9979 for a welding current of 220 A. Under the same fracture criteria, the maximum mean value of the correlation coefficient R was 0.997022 with the Swift-H-S combination model (λ = 0.2). Seen in this light, taking the maximum value of the correlation coefficient, the results achieved by using the Swift-Voce model (welding current of 100 A and 170 A) and the Swift-H-S model (welding current of 140 A and 220 A) were relatively better through contrastive analysis.

In Appendix A, it can be seen that the minimum MSE value was 0.255664 simulated with the Swift-H-S combination model under a welding current of 100 A and λ = 0.2. When the welding current was 140 A and λ = 0.2, the determined minimum MSE value was 0.042449 with the H-S-Voce combination model. When the welding current was 170 A and λ = 0.8, the MSE was the minimum value (MSE = 0.194321) with the H-S-Voce and Swift-H-S models. Furthermore, for a welding current of 220 A, the minimum MSE value was 0.028845 with the Swift-H-S combination model (λ = 0.8). Similarly, under the same fracture criteria, the obtained minimum MSE value was 0.135420 with the H-S-Voce combination model (λ = 0.2) for different welding currents. In summary, from this perspective of taking the minimum MSE value and mean value, the results were that the Swift-H-S model (welding current of 100 A and 220 A) and the H-S-Voce model (welding current of 140 A and 170 A) were relatively better.

Considering the aforementioned two factors and the fact that there was a tiny discrepancy between correlation coefficient values for the same welding current, the model was selected, which obtained more accurate results. Therefore, the adopted principle was that when the model corresponding to the correlation coefficient value was inconsistent with the selected model corresponding to MSE, the larger correlation coefficient value was selected to obtain the corresponding model based on the selected model’s match with the minimum value of MSE. The parameter value λ was also occasionally changed. Furthermore, the better models obtained were the Swift-H-S model (welding current of 100 A, λ = 0.2, and 220 A, λ = 0.8) and the H-S-Voce model (welding current of 140 A, λ = 0.2, and 170 A, λ = 0.2) through contrastive analysis. It could be seen from the stress nephogram that the achieved fracture position simulated with the Swift-H-S model and the H-S-Voce model combined with the failure model was greatly closer to the fracture position of the tensile test for T-welded joint specimens, and fracture positions were all near the weld, whether simulation with the better combination models or tensile experiment, as shown in Figure 14. Thereby, it was further confirmed that these results simulated with the Swift-H-S model and the H-S-Voce model were in good agreement with the tensile test results.

## 5. Conclusions

According to the welding deformation results of the T-joint thin-walled parts and the tensile test of a T-welded joint, the deformed tensile specimen models under different welding currents using different working hardening models were obtained to investigate their tensile properties and fracture behaviors through a combination of FE simulation and experiment. The accuracy of different hardening models was also evaluated in this work. The conclusions were drawn as follows:(1)The FE model of T-joint thin-walled parts and tensile test specimens for T-welded joints was built to investigate the welding deformation and tensile process behaviors under different welding currents. Seven fraction criteria, which were formed by multiply strain hardening models combined with fracture failure models, were employed to depict the tensile properties and fracture behaviors of a T-welded joint. They were very suitable for investigating the tensile behaviors in detail, but the obtained tensile property results simulated with combination models were in better agreement with the experiment results than the single model on the whole. In addition, the effect of the variation in coefficient value λ in the combination model on the simulation results was not greatly apparent under the same fracture criteria.(2)Compared with the experiment results, for the obtained yield strength, the maximum relative error value was not more than 20% for the obtained results simulated with a single model. For tensile strength, this error was below 10%, whether the simulated results were with a single model or a combination model, and sometimes the most accurate result was less than 1%.(3)The welding deformation increased with the increase in welding current. Compared to the results obtained with a larger welding current, the initial fracture strain decreased when the welding current increased, to a considerable extent. It was indirectly shown that as welding deformation increased, the initial fracture strain decreased. The initial fracture strain was deeply affected by welding current, whether employing the single model or the combination model.(4)The correlation coefficient, mean squared error, and results of the tensile test were comprehensively considered, and the different fracture criteria were evaluated. These results, simulated with the Swift-H-S model and the H-S-Voce combination model, were more precisely in agreement with the tensile test results.

In this work, the influence relationship between welding deformation, mechanical strength, and fracture behaviors of a T-welded joint is analyzed. However, there are some factors that are not considered in this work. In the future, the subsequent research will be concentrated on the microstructures and local mechanical properties of different zones (fusion zone, heat-affected zone of a T-welded joint, and base material) and their correlations under different welding currents. Since the tensile behaviors and mechanical properties of a T-welded joint are affected by welding process parameters, the optimization of welding process parameters is also taken into consideration to control heat input, material flow, and residual stress, thereby analyzing the damage variation of a T-welded joint. The effect of microstructure variation, welding defects, and thermal-mechanical couple on damage evolution and fracture behaviors from a multi-scale perspective will also be investigated, which will enhance the tensile properties, soundness, and reliability of the T-welded joint and provide a theoretical basis for improving the welding assembly quality and preventing the fracture failure of large-scale complex thin-walled structures under a thermal-vibration-load environment in the future.

## Figures and Tables

**Figure 1 materials-16-04864-f001:**
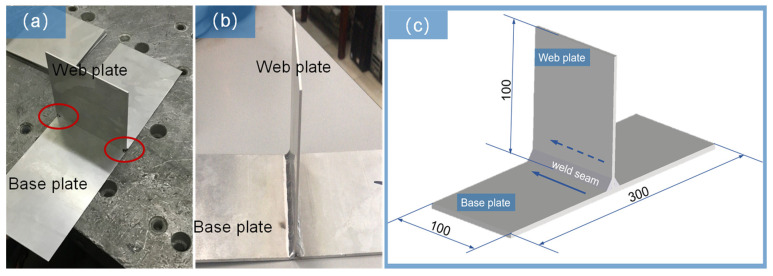
T-joint thin-walled part structure and size: (**a**) the fixed position of thin-walled part before welding assembly in the zone of red circle; (**b**) the T-joint thin-walled part after welding assembly; and (**c**) the structure size of the T-joint thin-walled part and welding direction marked with blue arrow.

**Figure 2 materials-16-04864-f002:**
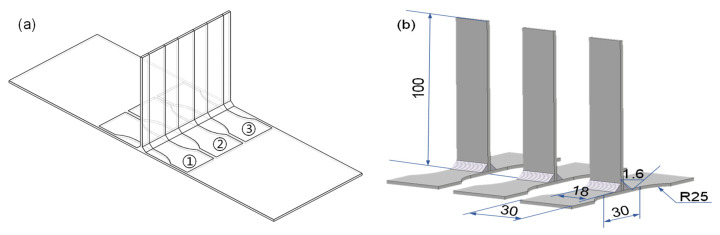
The structure of tensile specimens for T-welded joint: (**a**) the cutting position of the tensile specimen, where the numbers ①, ② and ③ denote the obtained tensile specimen and (**b**) the size of the tensile specimen.

**Figure 3 materials-16-04864-f003:**
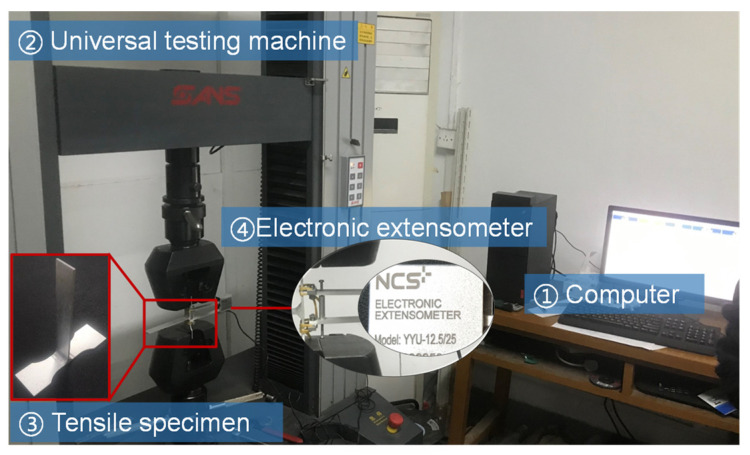
Experimental device for tensile tests. Where ①, ②, ③ and ④ respectively respresent the used computer, the SANS electronic universal testing machine, the used tensile specimen and electronic extensometer when the tensile test is carried out.

**Figure 4 materials-16-04864-f004:**
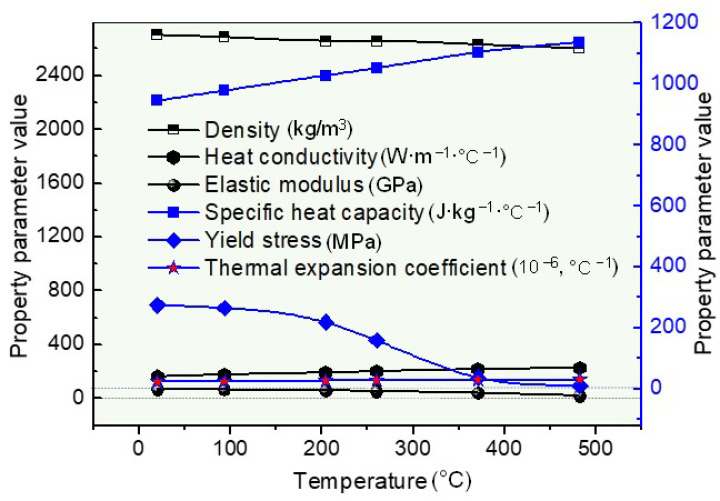
The property variation curve of AA6061-T6 with temperature.

**Figure 5 materials-16-04864-f005:**
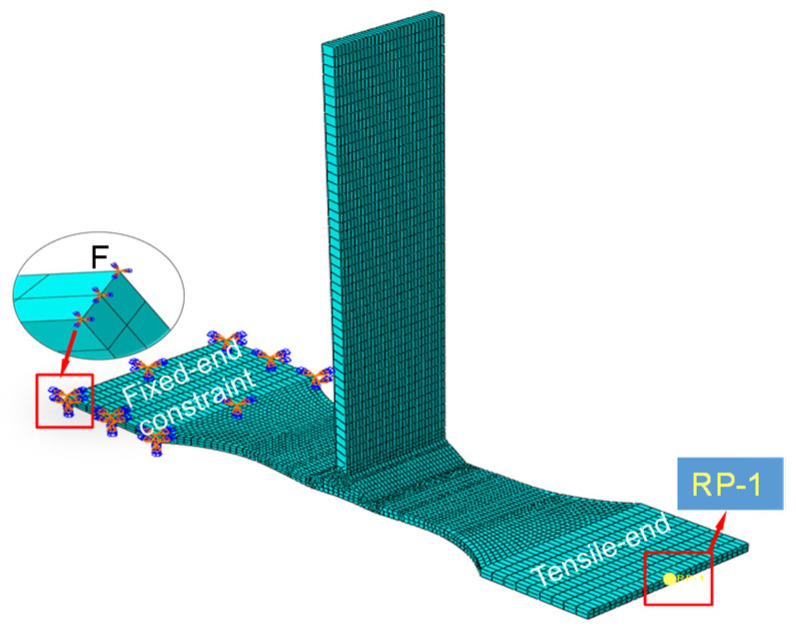
FE model and the constraint conditions of a tensile specimen. Where F denotes the fixed constraint of nodes, RP-1 represents the reference point of tensile test simulation.

**Figure 6 materials-16-04864-f006:**
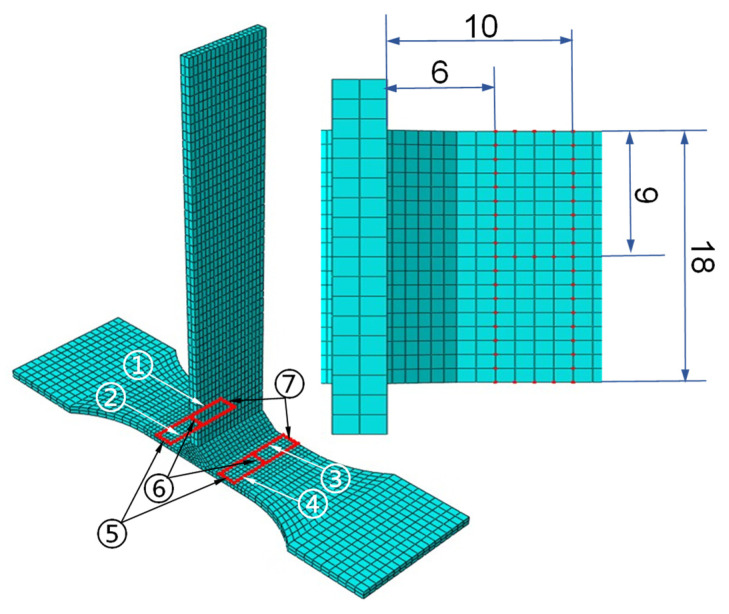
The different paths and sizes marked on the base plate of the tensile specimen model. Where the numbers ① to ⑦ denote the path-numbering of seven different paths.

**Figure 7 materials-16-04864-f007:**
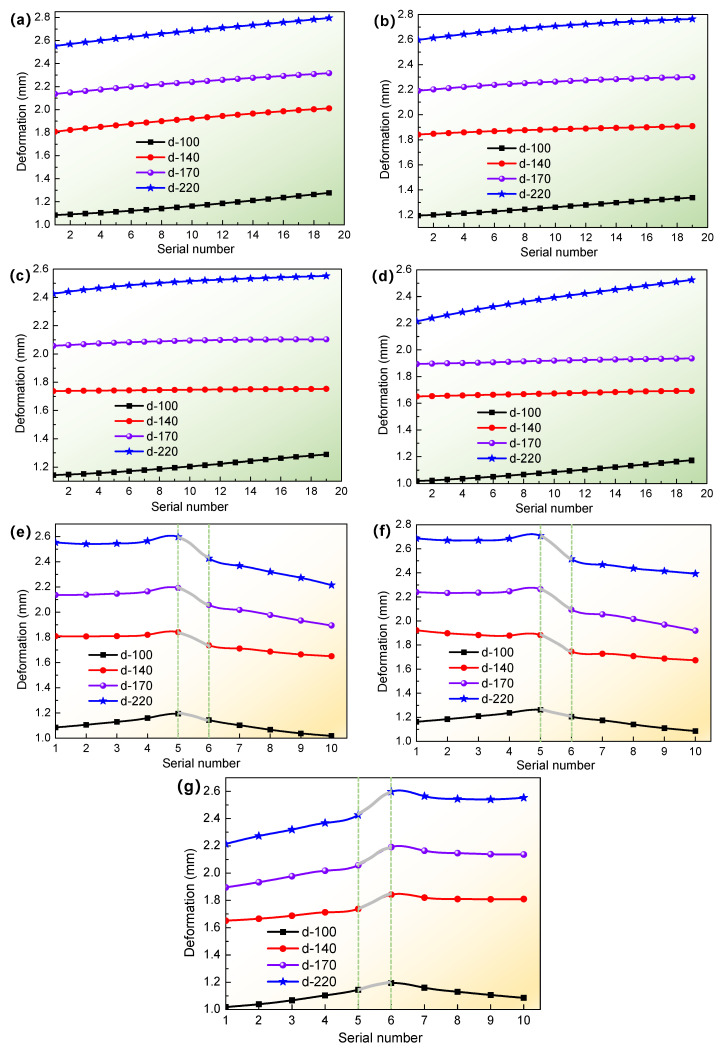
Deformation curves for different paths: (**a**) path 1, (**b**) path 2, (**c**) path 3, (**d**) path 4, (**e**) path 5, (**f**) path 6, and (**g**) path 7. Where d-100 denoted the deformation value when the welding current was 100 A, and the meaning of other symbols is similar.

**Figure 8 materials-16-04864-f008:**
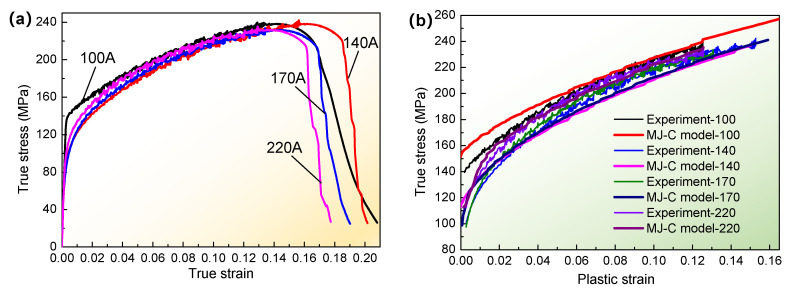
Stress-strain curves of tensile experiment and simulation through modified J-C model under different conditions: (**a**) true stress-strain curves through tensile test under different welding currents; (**b**) true stress-plastic strain curves for tensile experiment data and the modified J-C model, where experiment-100 denotes the true stress-plastic strain data, and MJ-C model-100 is the obtained data through modified and simplified Johnson-Cook model with FE simulation, when welding current was 100 A. The meaning of other symbols was similar.

**Figure 9 materials-16-04864-f009:**
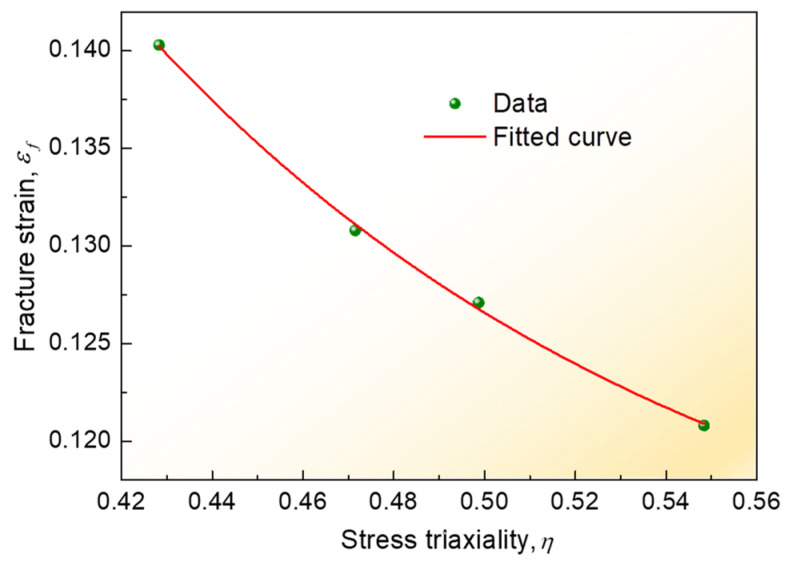
The relationship curve for stress trixiality-fracture strain.

**Figure 10 materials-16-04864-f010:**
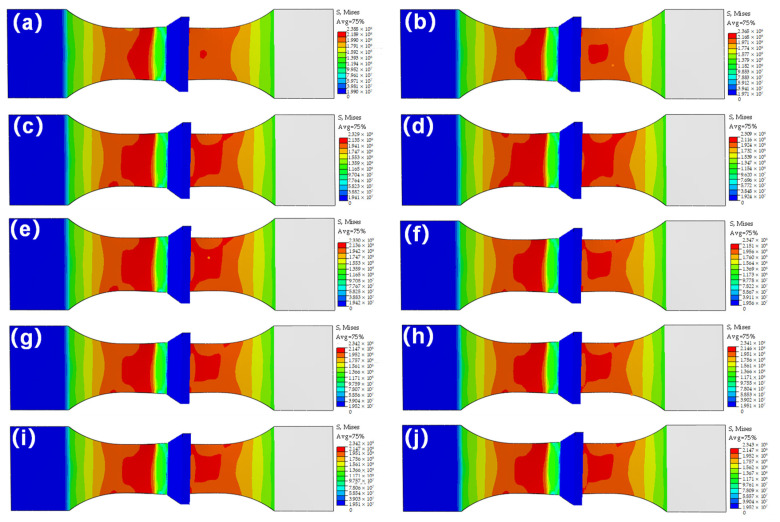
Mises stress nephogram with different models at a welding current of 220 A: (**a**) J-C model; (**b**) Swift model; (**c**) H-S model; (**d**) Voce model; (**e**) H-S-Voce combination model (λ = 0.2); (**f**) Swift-Voce combination model (λ = 0.2); (**g**) Swift-H-S combination model (λ = 0.2); (**h**) Swift-H-S combination model (λ = 0.4); (**i**) Swift-H-S combination model (λ = 0.6); (**j**) Swift-H-S combination model (λ = 0.8).

**Figure 11 materials-16-04864-f011:**
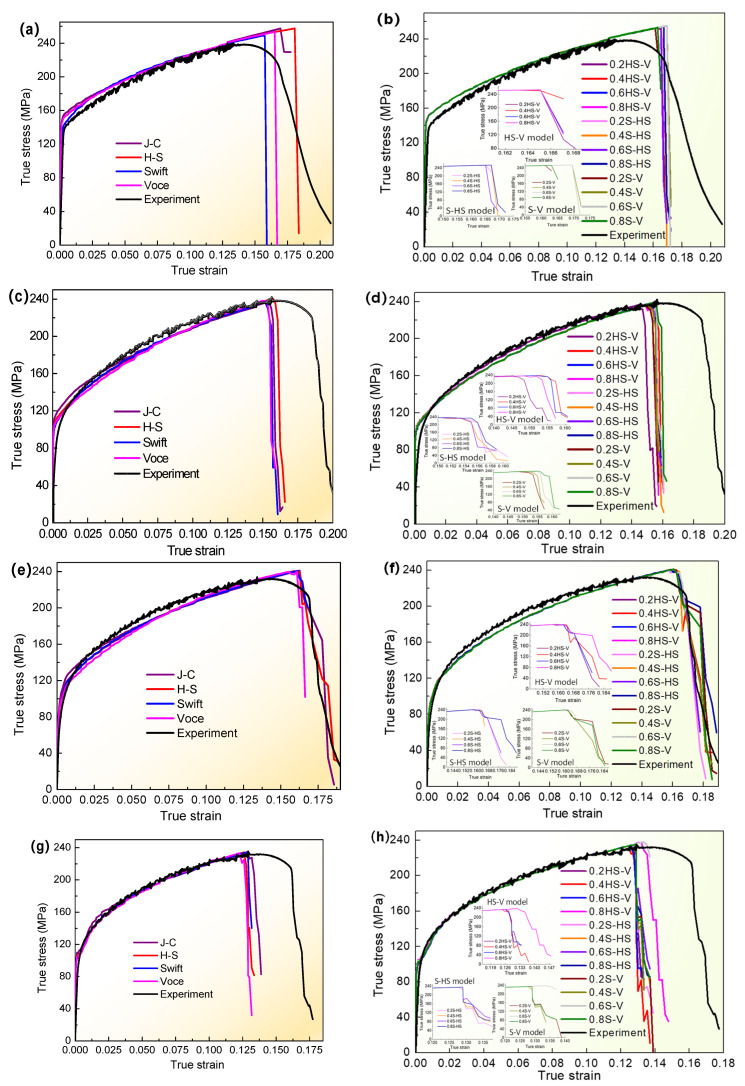
Comparison curves of simulated true stress-true strain with different models and tensile tests of T-welded joint specimens at different welding currents: (**a**,**b**) 100A; (**c**,**d**) 140 A; (**e**,**f**) 170 A; and (**g**,**h**) 220 A. Where J-C, H-S, Swift, Voce, and experiment, respectively, denoted the calculated and obtained true stress-strain data simulated with Johnson-Cook model, Hockett-Sherby model, Swift model, Voce model, and experiment, and 0.2HS-V, 0.4HS-V, 0.6HS-V, and 0.8HS-V, respectively, indicated the obtained true stress-strain data model simulated with H-S and Voce combination model when λ = 0.2, 0.4, 0.6, and 0.8. The meaning of other symbols is similar.

**Figure 12 materials-16-04864-f012:**
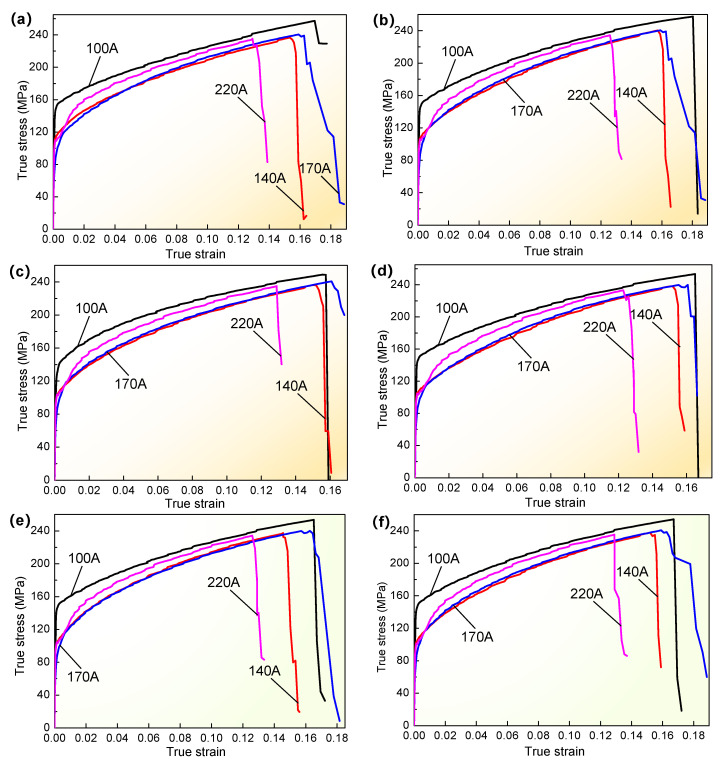
Comparison curves of simulated true stress-true strain under the same model at different welding currents: (**a**) J-C model; (**b**) H-S model; (**c**) Swift model; (**d**) Voce model; (**e**) H-S-Voce combination model, λ = 0.2; (**f**) Swift-H-S combination model, λ = 0.8.

**Figure 13 materials-16-04864-f013:**
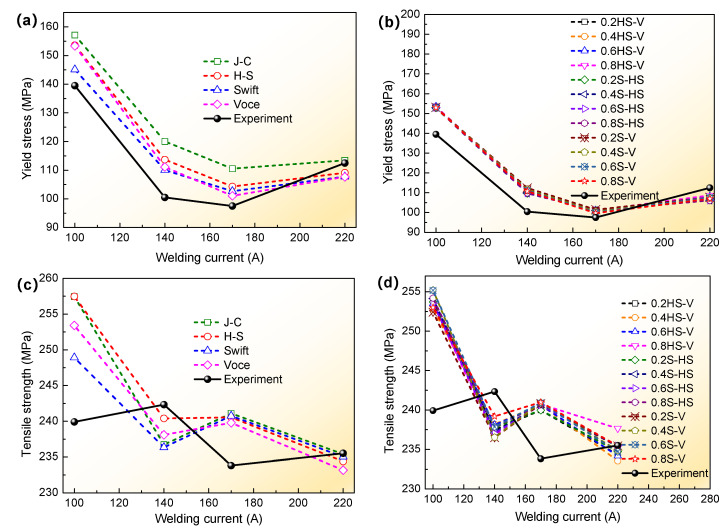
Variation trend of tensile properties simulated with different models and tensile tests at different welding currents: (**a**) the obtained yield stress variation trend using a single model and tensile test; (**b**) the obtained yield stress variation trend using a combination model; (**c**) the obtained tensile strength variation trend using a single model and tensile test; (**d**) the obtained tensile strength variation trend using a combination model.

**Figure 14 materials-16-04864-f014:**
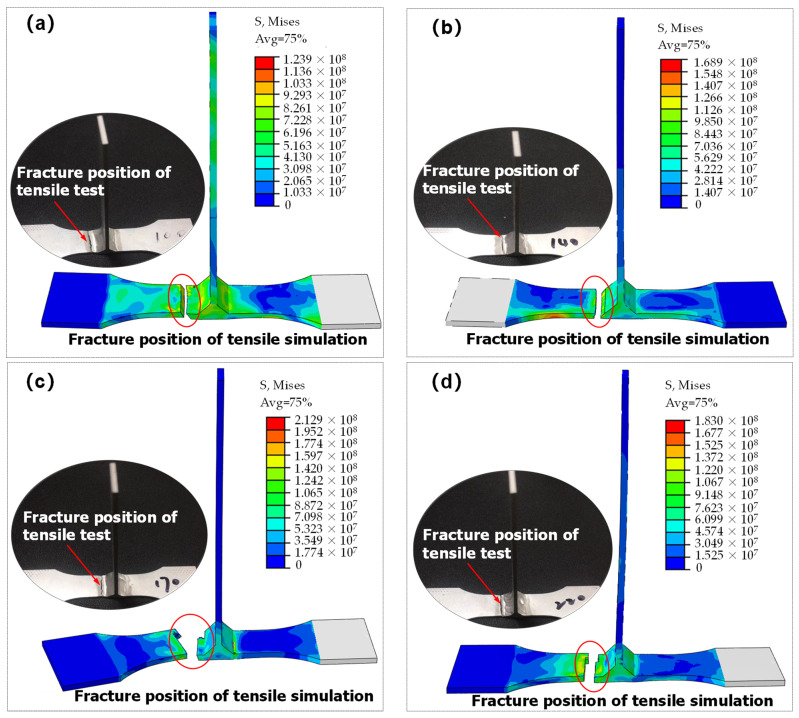
Stress nephogram with simulation and fracture position of tensile test for T-welded joint specimens: (**a**) the simulated result using Swift-H-S model when welding current is 100 A, λ = 0.2; (**b**) the simulated result using H-S-Voce model when welding current is 140 A, λ = 0.2; (**c**) the simulated result using H-S-Voce model when welding current is 170 A, λ = 0.2; (**d**) the simulated result using Swift-H-S model when welding current is 220 A, λ = 0.8.

**Table 1 materials-16-04864-t001:** The relevant parameter settings.

Stefan–Boltzmann Constant	Convective Heat Transfer Coefficient	Latent Heat	Solidus Temperature	Liquidus Temperature	Poisson’s Ratio
5.68 × 10^−8^	80 J/(m^2^·s·°C)	3.9 × 10^5^ J/kg	585 °C	659 °C	0.33

**Table 2 materials-16-04864-t002:** The number of mesh elements for different welding currents.

Welding-Deformed Model	100 A	140 A	170 A	220 A
Number of elements	18,863	17,358	12,389	18,633

## Data Availability

The data presented in this study are available in this article (tables and figures).

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
