# Peer review of "Tensile Behaviors and Mechanical Property Analyses of T-Welded Joint for Thin-Walled Parts in Consideration of Different TIG Welding Currents Using Multiple Damage Models and Fracture Criterions: Numerical Simulation and Experiment Validation"

_materials, 2023, doi:10.3390/ma16134864_

Round 1

Reviewer 1 Report

See the reviewer's comments file.

Reviewer 2 Report

The work presented by the authors titled "Tensile behaviors and mechanical properties analysis of T-welded Joint for thin-walled parts in consideration of different TIG welding currents using multiple damage models and fracture criterions: numerical simulation and experiment validation" is a very good work. There are some minor suggestions which will make the manuscript in more good shape.

1- The contributions of the work should be presented in the form of bullet points at the end of the introduction.

2- The results should be presented in Tabular form with comparison to the state of the art work.

3- Some more analysis on the results should be presented.

Reviewer 3 Report

The paper gives a detailed description of a new methodology to define the characteristic properties of T-welded joints obtained under different operative conditions.

The paper has to be modified by the authors to be reconsidered for the final publication. The following list of comments has to be addressed:

1. for those who are not familiar with the welding process in general, some additional details regarding the welding wire ER4043 would be of great help.

2. How did the authors choose the mentioned four levels of welding current?

3. Line 136: the sentence seems not to have an end.

4. Regarding figure 2: what was the extraction direction of the base plates? Parallel to RD?

5. Correct the label "Computure" in Figure 3.

6. It is not completely clear how the numerical simulations were run, at least what concerned the welding simulation: 1) the problem was solved according to a coupled temperature-displacement approach? 2) according to Figure 5, it seems that the welding simulation was carried out directly on the final geometry. The authors should better describe and justify their assumption. 3) on the contrary, if the welding simulation was carried out on the base geometries, how did the cut of the specimens was simulated?

7. Plots in Figure 8a are really difficult to be read. The suggestions is to provide a single numerical-experimental comparison for each of the investigated welding currents.

8. Figure 9 is placed after Figure 10. Please correct.

9. The authors mention the VUSDFLD subruotine: for the non-Abaqus user, some additional details would be of great help.

10. Why some curves in Figure 11 reaches the peak value and then they drop so sharply?

11. Looking at Figure 13, even though in some cases the percentage error can be considered limited, the trend predicted by the simulations is sometimes opposite than the real behavior as it can be seen in fFigure c and d.

12. Figure 14 can be reproposed in a more convenient way: for example it could be shown how much is the error between the position of the crack predicted by the simulations and the real location of rupture. 

Some editing languages as well as type errors should be carefully revised by the authors.

Reviewer 4 Report

The authors of the paper perform welding of AA6061, tensile tests, as well as simulation of welding and tensile tests. The experiment and simulation are performed at different current values to reveal the effect of current on stress and fracture. In my opinion, quite realistic simulation curves are encountered in the work. However, in the final analysis, the mismatch of 15-20 MPa between the simulated ultimate strength and the experimental one is a lot. Especially considering that a relatively simple alloy is used. 

There are no specific remarks, the work is performed at a good technical level, but I would say that the work lacks novelty and relevance. It might be worth clarifying this point. Also I would advise to verify the model on the original alloy (before welding).

Round 2

Reviewer 3 Report

The comments have been addressed satisfactorily